# Investigating *USP42* Mutation as Underlying Cause of Familial Non-Medullary Thyroid Carcinoma

**DOI:** 10.3390/ijms25031522

**Published:** 2024-01-26

**Authors:** Elisabete Teixeira, Cláudia Fernandes, Maria Bungărdean, Arnaud Da Cruz Paula, Raquel T. Lima, Rui Batista, João Vinagre, Manuel Sobrinho-Simões, Valdemar Máximo, Paula Soares

**Affiliations:** 1Cancer Signalling and Metabolism Group do Instituto de Investigação e Inovação em Saúde—i3s, Rua Alfredo Allen 208, 4200-135 Porto, Portugal; 2Cancer Signalling and Metabolism Group do Instituto de Patologia e Imunologia Molecular da Universidade do Porto—Ipatimup, Rua Júlio Amaral de Carvalho 45, 4200-135 Porto, Portugal; 3Departamento de Biomedicina da Faculdade de Medicina da Universidade do Porto—FMUP, Alameda Prof. Hernâni Monteiro, 4200-319 Porto, Portugal; 4Departamento de Bioquímica da Faculdade de Ciências da Universidade do Porto—FCUP, Rua do Campo Alegre 1021 1055, 4169-007 Porto, Portugal; 5Departamento de Patologia e Imunologia Molecular do Instituto de Ciências Biomédicas Abel Salazar da Universidade do Porto—ICBAS, R. Jorge de Viterbo Ferreira 228, 4050-313 Porto, Portugal; 6Department of Pathology, Iuliu Haţieganu University of Medicine and Pharmacy, Municipal Clinical Hospital, Cluj-Napoca 400139, Romania; 7Departamento de Patologia da Faculdade de Medicina da Universidade do Porto—FMUP, Alameda Prof. Hernâni Monteiro, 4200-319 Porto, Portugal

**Keywords:** hereditary cancer, familial thyroid cancer, Familial Non-Medullary Thyroid Carcinoma, molecular studies, ubiquitination

## Abstract

In a family with Familial Non-Medullary Thyroid Carcinoma (FNMTC), our investigation using Whole-Exome Sequencing (WES) uncovered a novel germline *USP42* mutation [*p.(Gly486Arg)*]. USP42 is known for regulating p53, cell cycle arrest, and apoptosis, and for being reported as overexpressed in breast and gastric cancer patients. Recently, a *USP13* missense mutation was described in FNMTC, suggesting a potential involvement in thyroid cancer. Aiming to explore the *USP42* mutation as an underlying cause of FNMTC, our team validated the mutation in blood and tissue samples from the family. Using immunohistochemistry, the expression of USP42, Caspase-3, and p53 was assessed. The *USP42* gene was silenced in human thyroid Nthy-Ori 3-1 cells using siRNAs. Subsequently, expression, viability, and morphological assays were conducted. p53, Cyclin D1, p21, and p27 proteins were evaluated by Western blot. USP42 protein was confirmed in all family members and was found to be overexpressed in tumor samples, along with an increased expression of p53 and cleaved Caspase-3. siRNA-mediated *USP42* downregulation in Nthy-Ori 3-1 cells resulted in reduced cell viability, morphological changes, and modifications in cell cycle-related proteins. Our results suggest a pivotal role of *USP42* mutation in thyroid cell biology, and this finding indicates that USP42 may serve as a new putative target in FNMTC.

## 1. Introduction

Thyroid cancer (TC) incidence has increased substantially over the last three decades [1]. Up to 10% of TC cases encompass familial forms [2]. When arising from thyroid follicular cells, familial cases of thyroid carcinoma are designated as Familial Non-Medullary Thyroid Carcinoma (FNMTC) [3]. FNMTCs are further subdivided into syndromic FNMTC (SFNMTC) and non-syndromic FNMTC (NSFNMTC). In SFNMTC, patients exhibit a predominance of non-thyroid tumors alongside thyroid cancer, and several genetic alterations associated with syndromic forms have been well described (e.g., APC, PTEN, and *DICER1*) [4,5]. NSFNMTC patients present with TC as the main clinical manifestation, comprising a very heterogeneous disorder, both at the phenotypic and genotypic levels, which is usually coupled with various low- to moderate-penetrance susceptibility alleles (e.g., FOXE1 and TITF-1), and incomplete penetrance [6,7]. A polygenic pattern was suggested but many questions remain unanswered [5,8,9].

Our team identified a family with FNMTC. The family was composed of a mother and two monozygotic twin daughters with an early-onset diagnosis of Oncocytic Subtype of Papillary Thyroid Carcinoma (OSPTC). The family fulfilled the criteria for hereditary cancer, but no alterations associated with syndromic forms of the disease were found. Using Whole-Exome Sequencing (WES), our team identified a possible disease-causative mutation in a Ubiquitin-Specific Protease gene [*USP42*
*p.(Gly486Arg)*].

Many USPs have been put forward as regulators of tumor formation and proliferation, mainly because they are involved in cell cycle progression, regulation of apoptosis-related factors, DNA repair activity, and other important cellular functions [10,11,12]. *USP42*, located on chromosome 7 (7p22.1), was shown to regulate cellular metabolism through p53–histone H2B interaction in cell lines by Hock et al. (2011) and Hock et al. (2014) [13,14].

To our knowledge, no prior studies investigating the role of *USP42* gene alterations in NSFNMTC have been performed; therefore, we decided to evaluate *USP42* mutation as an underlying cause of NSFNMTC.

## 2. Results

### 2.1. Case Report

Our team identified a family presenting with NSFNMTC. The family was composed of a mother (I), diagnosed at the age of 35, and of two monozygotic twin daughters both diagnosed at the age of 19 (II.1 and II.2) (Figure 1A). The histological lesions of the three cases were similar, all presenting the same diagnosis of OSPTC (Figure 1A). The mother, in addition to displaying OSPTC, also presented several lesions of Follicular Nodular Disease (FND) comprising oncocytic cells. FND lesions were also present in the surgical specimens from the twins.

### 2.2. Molecular Studies

#### 2.2.1. *USP42* Missense Mutation Validation

In silico analysis of the normal DNA extracted from the three affected individuals was subjected to WES (Figure 1A,B), which revealed the presence of an extremely rare Single-Nucleotide Polymorphisms (SNP) affecting the *USP42* gene (Table 1). Although this is a variant of unknown significance according to ClinVar, it shows a predicted CADD score over 20, meaning that this SNP is predicted to be a damaging substitution (Table 1). After confirming the presence of this SNP on IGV (Integrative Genomics Viewer), PCR followed by Sanger sequencing was performed using DNA from blood samples and from FFPE (Formalin-Fixed Paraffin-Embedded) samples of tumoral and non-tumoral tissues (Figure 1C,D; Table 1). The *USP42*
*p.(Gly486Arg)* mutation was present in all analyzed samples of the mother (I, proband) and of the monozygotic twin daughters (II.1 and II.2) in heterozygosity (Figure 1D; Appendix A).

When looking at mutations affecting the *USP42* gene in 514 PTCs subjected to WES that were selected from the TCGA database, two missense mutations were identified, one of which is pathogenic [*USP42*
*p.(Pro669Gln)*] (Figure 1E; Appendix A). We expanded our analysis and looked at PTC cases harboring pathogenic somatic alterations affecting any genes of the USP family. Of the 17 PTC cases found to have genetic alterations in USPs, 24% and 16% had mutations affecting *USP9X* and *USP6*, respectively. In addition, 76% of these 17 PTCs also harbored *BRAF* hotspot mutations (Appendix A; Appendix A).

The tumor sample from the mother subjected to WES had a similar variant-allele frequency of the *USP42*
*p.(Gly486Arg)* SNP to the one observed in the matched normal sample (Figure 1F). CNA analysis revealed only a loss of chromosome 22 (Figure 1G). When looking at the somatic mutations, an *HRAS p.(Gln61Arg)* hotspot mutation was identified (Figure 1H; Table 2).

#### 2.2.2. Immunohistochemistry

FFPE tissues from the affected individuals were assessed to observe tissue expression of USP42. We observed that USP42 protein was overexpressed in all tumoral tissues from all tested individuals (Figure 2B,D,F). Adjacent tissues also presented high expression of the protein, thus suggesting that *USP42* mutation may result in protein overexpression (Figure 2A,C,E).

As for the other markers analyzed, we observed that Caspase-3 (Figure 3) and p53 (Figure 4) proteins presented tumor overexpression in all tested samples.

Quantification of p53 showed that the protein was overexpressed in the tumor tissues compared to adjacent tissues and to non-tumoral tissue from the proband (2.71% p53 positive cells) (Table 3; Figure 4). Tumor lesions of the three elements presented a range of 26.33–57.56% p53-positive cells in the available fragments, while adjacent tissue presented 0.75–10.65% p53-positive cells in the available fragments (Table 3; Figure 4).

#### 2.2.3. *USP42* Silencing and Analysis of mRNA and Protein Expression 

We successfully silenced *USP42* using two different siRNA treatments, siRNA 2 and siRNA 7, which were evaluated for mRNA expression using RT-qPCR with the 2^−ΔΔCt^ method (Figure 5A). At the 24 h timepoint, siRNA 2 and siRNA 7 treatments each had a significant silencing effect on the gene (60%) when compared to a negative control (*p* < 0.0001 for both treatments). At the 48 h timepoint, the silencing effect (60%) was maintained for both treatments (*p* = 0.0001 and *p* = 0.0003, respectively). However, at 72 h, a recuperation of *USP42* gene expression was observed, but silencing effects of 51% and 45% were still detected for siRNA 2 (*p* = 0.0002) and siRNA 7 (*p* = 0.004), respectively (Figure 5A). USP42 protein expression was evaluated by Western blot in these same treatments. We did not observe protein expression alterations at 24 h or 48 h timepoints (Figure 5B). However, at the 72 h timepoint, USP42 protein silencing was significant using the siRNA 7 treatment, with a 32% reduction in protein expression compared to the negative control (*p* = 0.008). No significant changes were observed for protein expression using siRNA 2 for up to 72 h post-treatment.

#### 2.2.4. Cell Viability and Apoptosis Analysis

We performed flow cytometry analysis with Annexin V/PI double staining to measure cell viability/mortality following siRNA treatment (Figure 6; Appendix A). At the 24 h timepoint, the level of viable cells (Annexin V^−^/PI^−^) showed a statistically significant decrease when treated with siRNA 2 compared to the negative control (*p* = 0.032) (Figure 6A). No statistically significant changes were found between treatments at the 48 h timepoint. However, after 72 h of treatment cell viability levels decreased in cells treated with siRNA 7, compared to negative control (*p* = 0.010) (Figure 6A). It was also observed that siRNA 7 treatment resulted in lower cell viability compared to siRNA 2 treatment (*p* = 0.013) (Figure 6A).

Necrotic cells (Annexin V^−^/PI^+^) represented the smallest population of cells. No statistically significant differences were observed between treatments or timepoints (Figure 6B). For the population of early apoptotic cells (Annexin V^+^/PI^−^), at the 24 h timepoint siRNA 2 treatment seemed to result in a higher frequency of early apoptotic cells compared to siRNA 7 treatment (*p* = 0.023) (Figure 6C). However, at the 48 h timepoint, siRNA 7 treatment showed to be more effective when compared to both negative control and siRNA 2 treatments (*p* < 0.0001, for both comparisons) (Figure 6C). The same observation was made at the 72 h timepoint (*p* = 0.004 and *p* = 0.005, respectively) (Figure 6C). Late apoptotic cells (Annexin V^+^/PI^+^) were the second most representative population of cells present in the samples (Figure 6D). The frequency of late apoptotic cells was higher following siRNA 2 treatment at the 24 h timepoint than in the negative control (*p* = 0.027) (Figure 6D). At 48 h, no differences were observed (Figure 6D). At the 72 h timepoint, siRNA 7 treatment resulted in higher percentages of late apoptotic cells compared to negative control and siRNA 2 treatment (*p* = 0.032 and *p* = 0.037, respectively) (Figure 6D).

#### 2.2.5. *USP42* Silencing Effect in Cell Proliferation

To assess if cell proliferation was affected by siRNA treatments, we performed cell counting with Trypan Blue on a Neubauer chamber. No statistically significant changes were found at 24 h and 48 h for both siRNA treatments, although a tendency for decreased proliferation was found when treating cells with siRNA 7 at 48 h (*p* = 0.065) (Figure 7). However, at the 72 h timepoint, siRNA 7-treated cells were the most affected, showing lower proliferation compared to negative control cells (*p* = 0.0002) and siRNA 2-treated cells (*p* = 0.014) (Figure 7). At the 72 h timepoint, compared to the negative control siRNA 2-treated cells proliferated two times less and siRNA 7-treated cells proliferated 7.3 times less (Figure 7).

#### 2.2.6. Cell Morphology Evaluation by Phalloidin Staining

Cell morphology was evaluated by immunofluorescence following 48 h treatment with siRNAs. Representative images of each condition are shown in Figure 8. Negative control-treated cells presented, as expected, an epithelial-like phenotype, higher cellular confluence, and cell-to-cell adhesion, with actin filaments being evenly distributed within the cells (Figure 8A,B). For both treatments, siRNA 2 and siRNA 7, most of the fields were composed of a significant number of cells with altered morphology. These cells were smaller and their actin filaments appeared to lose shape in the cytoplasm and to be concentrated in the cell periphery (Figure 8C–F). The described effect was more evident with siRNA 2 treatment, which presented a higher number of cells with altered morphology per field at the 48 h timepoint (Figure 8).

#### 2.2.7. β-Actin Expression Analysis by Western Blot

To assess whether β-actin alterations were only morphological or whether siRNA treatment resulted in β-actin protein expression changes, we performed an evaluation using Western blot. At 24 h, no evident changes in β-actin expression were found. However, at 48 h we observed a reduction of 55% in β-actin expression for siRNA 2 treatment compared to the negative control (*p* = 0.002) (Figure 9). A reduction of 53% in β-actin expression was also verified using siRNA 7 treatment, although this was not statistically significant (*p* = 0.065) (Figure 9). At the 72 h timepoint, no difference was found between β-actin expression in siRNA 2-treated cells and negative control cells; however, a significant 48% reduction was found in siRNA 7-treated cells (*p* = 0.010) (Figure 9). We also detected reductions in tubulin and vinculin expression by Western blot (Appendix A).

#### 2.2.8. Cell Cycle-Related Proteins Analyzed by Western Blot

As USP42 has been described to regulate cell cycle and apoptosis [13], protein expression of cell cycle-related proteins (p53, Cyclin D1, p27, and p21) was evaluated (Figure 10).

siRNAs had not altered USP42 protein expression at 48 h post-treatment; however, p53 expression was decreased at this timepoint (Figure 10A). Compared to the negative control, siRNA 2 treatment led to a 40% reduction in p53 expression (*p* = 0.004) and siRNA 7 treatment lead to a 53% reduction (*p* = 0.037) (Figure 10A). After 72 h, siRNA 2 treatment did not give a statistically significant change compared to the negative control, but p53 expression in siRNA 2-treated cells presented a large discrepancy compared to siRNA 7-treated cells (*p* = 0.002) (Figure 10A). At the 72 h timepoint, we observed the most significant reduction in p53 expression (66%) after treating cells with siRNA 7 (*p* < 0001) (Figure 10A).

Cyclin D1 expression was only found to be significantly altered at the 24 h timepoint, with a 38% reduction in its expression following siRNA 7 treatment compared to negative control (Figure 10B). No significant changes were observed at the remaining timepoints (Figure 10B).

p21 protein expression was reduced following siRNA 2 treatment. At 24 h, we observed a 27% decrease of expression compared to negative control (*p* = 0.033) and at the 48 h timepoint, a 58% reduction was found (*p* = 0.0004) (Figure 10C). However, at the 72 h timepoint with siRNA 2 treatment, cells seemed to recuperate p21 expression to the same level as in negative control cells (Figure 10C). Using siRNA 7, only at the 48 h timepoint a statistically significant association was found, interestingly with a marked increase of 140% (2.4-fold) of p21 expression compared to negative control (*p* = 0.045) (Figure 10C). When comparing p21 expression between the two treatments, a 2-fold difference was found (*p* = 0.011) (Figure 10C).

When treating cells with siRNA 2, we observed a reduction in p27 expression at the 24 h timepoint (33% reduction, *p* = 0.003) and at the 48 h timepoint (51% reduction, *p* = 0.007) (Figure 10D). As for siRNA 7 treatment, we found a statistically significant increase of 39% on p27 expression at the 72 h timepoint (*p* = 0.014) (Figure 10D).

## 3. Discussion

We report here the identification of a *USP42* germline mutation that segregated with the disease in a family compatible with the diagnosis of NSFNMTC. The family was composed of a mother and two monozygotic twin daughters with an early-onset diagnosis of OSPTC. Despite filling the criteria for hereditary cancer, no alterations associated with syndromic forms of FNMTC were found. Using WES, we identified a possible causative mutation for the disease in a Ubiquitin-Specific Protease gene-[*USP42*
*p.(Gly486Arg)*].

USP42 is a thiol-dependent deubiquitinating enzyme (DUB) predicted to be active in cytosol and nucleus [15,16]. The ubiquitination pathway is a dynamic and complex post-transcriptional process, regulated by E3 enzymes and reversible through DUBs [11,17,18]. DUBs are enzymes capable of removing ubiquitin from ubiquitinated proteins, which may regulate their stability and activity, protecting them from proteasomal degradation [19,20,21].

*USP42* gene alterations have been previously described as an underlying cause of cancer. Wang et al. (2022) suggested that *USP42* might be a target gene for breast cancer [22]. They have shown that USP42 was overexpressed in breast cancer tissues and further demonstrated, using in vitro studies, that USP42 overexpression promoted cell invasion and migration, as well as that *USP42* silencing led not only to a reduction in proliferation, but also to increased apoptosis, suggesting that this gene may act as an oncogene [22]. USP42 regulation of cellular metabolism through p53—histone H2B interaction has been demonstrated [13,14].

Other USPs have been put forward as regulators of tumor formation and proliferation in TC. Jianing et al. (2022) showed an interaction of USP26 protein with TAZ protein, leading to its stabilization and the consequent proliferation of Anaplastic Thyroid Carcinoma (ATC) [23]. Liang et al. (2021) reported that TC patients with low expression of USP18 had worse survival rates than those with high USP18 expression [24]. An et al. (2015) demonstrated that USP39 underexpression was associated with decreased cell proliferation due to G2/M cell cycle arrest in Medullary Thyroid Carcinoma (MTC) cells [25]. More recently, using WES, Maria et al. (2022) described a *USP13*
*p.(Val495Met)* mutation as an underlying cause of PTC in a family with NSFNMTC. In this family, the *USP13* mutation that was shown to result in USP13 overexpression, which played a role in the tumorigenesis of PTC [26].

These results suggest that USPs may play a role in both sporadic and familial thyroid carcinoma. We examined the data available in the TCGA database and found that 17 PTC cases had somatic genetic alterations in USP genes. *USP9X* and *USP6* were the most frequently affected USP genes. Two cases presented variants in *USP42* (different from the one identified in the family examined in this study). *BRAF* hotspot mutations were present in 76% of these 17 PTCs, suggesting that USP alterations can cooperate with well-known TC genetic alterations during the transformation process. Recently, a polygenic mode of action in NSFNMTC tumorigenesis has been proposed [8,9]. NSFNMTC is described as presenting low- to moderate-penetrance susceptibility alleles with incomplete penetrance, which leads to TC susceptibility [5,6,7]. It is hypothesized that the combined influences of other modified genes along with these susceptibility alleles triggers the onset of cancer [6,7,8,9]. Of note, in the studied family we detected a somatic *HRAS* mutation in the proband tumor samples, but that alteration was not present in the other elements.

The *USP42*
*p.(Gly486Arg)* mutation was present in all of the tumor samples from the family members in heterozygosity. Using immunohistochemistry (IHC) we noted that, although USP42 was expressed in the normal tissue, an increased expression was observed in the tumor specimens, indicating overexpression. The tumor samples also presented higher expression of p53 and Caspase-3 than adjacent tissues.

To our knowledge, no prior studies investigating the role of *USP42* gene alterations in NSFNMTC have conducted, and we therefore performed several in vitro experiments to validate *USP42* mutation as an underlying cause of NSFNMTC.

We undertook downregulation of USP42 using two siRNAs, both of which efficiently silenced *USP42* at 24 h and 48 h, also showing some recuperation of the silencing effect at 72 h. A marked effect on apoptosis was shown, with a significant increase in apoptosis following silencing of the *USP42* gene. These results are consistent with those of Wang et al. (2022), who silenced *USP42* and observed increased apoptosis in breast cancer cells [22]. Other USPs, such as *USP15*, have been demonstrated to impact the stability and activity of Caspase-3 [27].

The effect of *USP42* siRNA silencing on proliferation was delayed and not so marked, although a significant reduction in the proliferation of *USP42*-silenced cells at 72 h post-treatment was noted.

The ubiquitination process is a dynamic process that has already been described to regulate actin filaments in the cells [28]. In our study, we observed that *USP42* silencing led to cytoskeletal alterations in the cells, as shown by phalloidin staining of actin filaments, and also a reduction in cytoskeletal proteins as shown by Western blot for actin, vinculin, and tubulin. Whether *USP42* silencing has a direct or indirect effect on cytoskeleton formation is still to be uncovered, but other USPs, such as *USP8* and *USP14*, have been shown to regulate cytoskeletal proteins [28,29].

It has been described that USP42 stabilizes p53 by removing ubiquitin and preventing its proteasomal degradation [13,14,30]. Using IHC, increased expression of p53 was found in FFPE tissues for all family members. Usually, increased p53 expression results from mutations [31], but no p53 mutations were found in the samples from the proband (I). Therefore, we hypothesize that increased p53 expression in the tissues is related to USP42 overexpression, which was also noted in the FFPE; supporting this hypothesis, after silencing *USP42* we observed a significant reduction in p53 expression at 48 h post-treatment for both siRNAs. However, at 72 h post-transfection, while siRNA 7 treatment maintained a significant reduction in p53, this effect was not verified for siRNA 2. This might be due to the transient effects of the siRNAs, which presented a recovery at 72 h, as referred to above. As the *USP42* gene can present different isoforms, we can also hypothesize that different siRNAs may target different isoforms, thus resulting in variable effects in the cell [13].

Hou et al. (2016) has shown that USP42 was overexpressed in gastric cancer patients and that overall survival of these patients was reduced compared to patients whose tumors presented lower expression of USP42 protein [32]. They further demonstrated, using siRNAs against the *USP42* gene, that cell cycle-related proteins, namely Cyclin D1, Cyclin E1, and the Proliferating Cell Nuclear Antigen (PCNA), were downregulated following *USP42* silencing [32]. Furthermore, Maria et al. (2022) showed that *USP13* silencing by siRNAs decreased expression of Cyclin D1, leading to cell cycle arrest [26]. In our study, in concordance with the aforementioned studies, we demonstrated that, following *USP42* silencing using siRNA 2, Cyclin D1 is significantly downregulated at 24 h post-treatment.

We also showed that p21 is downregulated at 24 h and 48 h after treating cells with siRNA 2, although the same effect is not evident when treating cells with siRNA 7 for 48 h. These discrepancies may be associated with the specific binding site of the siRNA that might lead to variable effects in the cells or they may be related to the cytotoxicity associated with siRNA 7. Hock et al. (2014) reported that *USP42* silencing resulted in lower expression of p21 mRNA and hypothesized that USP42 is recruited to the transcriptional site of p21, but further investigation on this interaction is still required [14]. A similar effect was obtained for p27 expression following *USP42*-silencing treatments. siRNA 2 treatment resulted in p27 downregulation between 24 h and 48 h of treatment, and siRNA 7 treatment resulted in the upregulation of p27 at the 72 h timepoint.

Our results, along with the available information in the literature, leads us to hypothesize that the *USP42*
*p.(Gly486Arg)* missense mutation is the underlying cause of NSFNMTC in this family. The mutation seems to be activating, leading to overexpression of the mutated protein, thus possibly acting as an oncogene.

As for the limitations of this study, it is important to emphasize that our sample is small, being composed of only three affected elements. Furthermore, in the in vitro siRNA studies we cannot rule out the possibility that some of the observed changes might be due to the transient nature of siRNA and/or that some effects may be due to transfection cytotoxicity. Future assays using CRISPR-Cas9 or a similar methodology, in which the Nthy-Ori 3-1 cell line will be transformed to harbor the specific SNP that was found in this family by using WES [*USP42*
*p.(Gly486Arg)*], will be necessary.

The genetics behind NSFNMTC is rather heterogeneous and poorly understood [2,4]. The causative genes in many affected families remain to be identified and the establishment of molecular markers would lead to better genetic counselling and to the development of new therapies. Disease prevention and active surveillance are key factors for reducing aggressive treatments, high comorbidities, and invasive surgeries, in addition to avoiding high burdens for the health care system.

## 4. Materials and Methods

### 4.1. Patients

The surgical procedures and sample collection were performed at Cluj-Napoca Municipal Clinical Hospital, Romania. Peripheral blood and FFPE tumor tissues were collected from the three family members. We obtained a total of 26 FFPE blocks from the three elements: 10 blocks from the mother (I), 12 from the first twin (II.1), and 4 from the second twin (II.2) (Figure 1A). DNA was extracted from all samples and preserved at −20 °C until further processing was performed. All subjects gave their informed consent for inclusion before they participated in the study. This study was conducted in accordance with the Declaration of Helsinki.

### 4.2. Molecular Studies

#### 4.2.1. DNA Extraction from Patient Samples

DNA was extracted from peripheral blood samples using the GRS Total DNA Kit—Blood & Cultured Cells (GRiSP Research Solutions, Porto, Portugal). DNA extraction from FFPE tissues was performed using the GRS Genomic DNA Kit BroadRange (GRiSP Research Solutions, Porto, Portugal), following the manufacturer’s instructions. Quantitative and qualitative sample analysis was performed using spectrophotometry with a Nanodrop N-1000 Spectrophotometer for microvolume UV–Visible measurements (Thermo Scientific, Waltham, MA, USA).

#### 4.2.2. Immunohistochemistry

IHC was performed in FFPE samples of the three family members using the Ultravision Quanto Detection System HRP (Epredia^®^, Kalamazoo, MI, USA, TL-125-QHL) following the manufacturer’s instructions. Dewaxing and antigen retrieval were performed using Dewax and HIER (Epredia, Kalamazoo, MI, USA) in a steamer for 45 min. Monoclonal USP42 (1:250, d-4 mouse, sc-390604, Santa Cruz Biotechnology, Dallas, Texas, USA), monoclonal Caspase-3 (1:100, E-8, sc-7272, Santa Cruz Biotechnology, Dallas, Texas, USA), and monoclonal p53 (1:250, mouse, NCL-L-p53-DO7, Leica, Chicago, Illinois) antibodies were used as primary antibodies. Detection was performed with a DAB chromogen for all antibodies (Epedria^®^, TA-125-QHDX, Kalamazoo, MI, USA). Slides were counterstained with Gill’s hematoxylin. USP42 and Caspase-3 staining were evaluated by semi-quantitative analysis. Quantification of p53 expression was performed by counting at least 1000 cells per field ratio between positive cells and total cell number was calculated as a percentage.

### 4.3. In Silico Studies

#### 4.3.1. Variant Calling and Annotation

Normal DNA was extracted from the blood of the three affected individuals, and the tumor DNA from the mother was microdissected from FFPE tissue blocks as previously described. All DNA samples were subjected to WES using the Illumina DNA Prep.

#### 4.3.2. Whole-Exome Sequencing

The resulting sequencing files were aligned against the human GRCh37 reference genome using the Burrows-Wheeler Aligner (BWA, v0.7.10) [33]. PCR duplicates were removed using Picard (http://broadinstitute.github.io/picard/ (accessed on 29 October 2023)). Single/nucleotide variants (SNVs) and insertions and deletions (indels) were annotated using GATK (version 3.1.1, Broad Institute, Cambridge, MA, USA) [34]. Reads around known and detected indels were realigned, and base quality was recalibrated using GATK (version 3.1.1, Broad Institute, Cambridge, MA, USA) [34].

All generated files were analyzed using The Cancer-Related Analysis of Variants Toolkit (CRAVAT (version 5.2.4, Johns Hopkins, Baltimore, MD, USA), Figure 1B) [35].

#### 4.3.3. Variant Prioritization

A step-by-step approach was used to identify potential pathogenic germline variants, using the following criteria: (i) having an allele fraction ≤1% in the overall population, as given by gnomAD (version 2.1.1) [36,37], (ii) being present in all three affected individuals, (iii) having a combined annotation dependent depletion (CADD (version 1.7, University of Washington, Hudson-Alpha Institute for Biotechnology, WA, USA and Berlin Institute of Health at Charité - Universitätsmedizin Berlin, Germany) [38] PHRED score 10), and (iv) being present on IGV(version 2.16.2 Broad Institute, Cambridge and Harvard, MA, USA) [39] when visualizing the sequenced reads. Shared candidate variants were subjected to confirmation by Sanger sequencing.

#### 4.3.4. Analysis of Somatic Variants in the Tumor DNA

The presence of somatic SNVs and indels in the tumor DNA of the mother were obtained by filtering all germline variants present in the matched normal DNA and by visualizing the sequenced reads of each somatic variant on IGV (version 2.16.2, Broad Institute, Cambridge and Harvard, MA, USA) [39]. Mutational hotspots were annotated according to Chang et al. (2016) [40]. FACETS (version 0.5.6, Memorial Sloan-Kettering Cancer Center, New York, NY, USA) [41] was used to determine Copy Number Alterations (CNAs) and whether genes harboring somatic or germline mutations were targeted by Loss Of Heterozygosity (LOH), as previously described [42]. The cancer cell fractions of all somatic mutations were computed using ABSOLUTE (version 1.0.6, Broad Institute, Cambridge and Harvard, MA, USA) [43]. A mutation was classified as clonal if its probability of being clonal was >50% or if the lower bound of the 95% confidence interval of its Cancer Cell Fraction (CCF) was >90%, as previously described [42].

#### 4.3.5. Polymerase Chain Reaction and Sanger Sequencing Analysis

The candidate germline variant was validated by Polymerase Chain Reaction (PCR) using the QIAGEN multiplex PCR kit (QIAGEN, Hilden, Germany) and by Sanger sequencing analysis, as previously described [44]. Primer design was performed (Forward: *5′-CAGCCAAAATGGGTTCTGTT-3′*; Reverse: *5′-TTTGGACAGAAGCTGAAGCA-3′*) (Integrated DNA Technologies, Coralville, IA, USA). The annealing temperature of 52 °C was established after protocol optimization for the *USP42* gene. All alterations were confirmed by performing a new and independent analysis.

#### 4.3.6. USP Genetic Alterations in Papillary Thyroid Cancers

Bioinformatic analysis was conducted on The Cancer Genome Atlas (TCGA) database (https://portal.gdc.cancer.gov/ accessed on 15 November 2023)). USP mutations, gene amplifications, gene deletions, frameshifts, truncations, and splice variants were searched. Other online databases describing *USP42* mutation status and effect prediction were searched, namely IntOGen (version 2.4.0) [45], gnomAD (version 2.1.1) [37] COSMIC (version 99) [46], Ensembl (version 111) [47] and ClinVar (version 1.0) [48].

### 4.4. Functional Characterization

#### 4.4.1. Cell Culture

Nthy-Ori 3-1 cells were cultured in T75 flasks (TPP Techno Plastic Products AG, Trasadingen, Switzerland) in RPMI-1640 medium (Capricorn scientific, Ebsdorfergrund, Germany), supplemented with 10% heat-inactivated Fetal Bovine Serum (Gibco, Thermo Fisher Scientific, Waltham, MA, USA) and 1% Penicillin/Streptomycin (Gibco, Thermo Fisher Scientific, Waltham, MA, USA). Cells were maintained at 37 °C with 5% CO_2_ humidified environment in an incubator (HERACell 150i, Thermo Fisher Scientific, Waltham, MA, USA). Cells were routinely screened to confirm the absence of mycoplasma contamination.

#### 4.4.2. siRNAs–*USP42* Silencing

Nthy-Ori 3-1 cells were plated in 6-well plates (1 × 10^5^ cells/well). After 24 h, cells were transfected with pre-designed siRNAs (Hs_USP42_2–SI00758618 and Hs_USP42_7–SI03068583; QIAGEN, Aarhus, Denmark) or with a non-coding siRNA (Allstars Negative Control siRNA-SI03650318, QIAGEN, Aarhus, Denmark) using Lipofectamine 2000 transfection reagent (Invitrogen, Thermo Fisher Scientific, Waltham, MA, USA) diluted in Opti-MEM™ I Reduced Serum Medium (Gibco, Thermo Fisher, Waltham, MA, USA). siRNA complexes were formed for 20 min in the dark and were added to the cells at the concentration of 25 nM. Medium change was performed 4h post-transfection (t = 0 h). Each experiment was performed with two replicates and at least three independent experiments were conducted. Cells were collected at 24 h, 48 h, and 72 h timepoints and counted with Trypan Blue (T8154, Merck KGaA, Darmstadt, Germany) on a Neubauer chamber for proliferation assessment.

#### 4.4.3. RNA Extraction and cDNA Conversion

RNA was extracted using the GRS Total RNA Kit–Blood & Cultured Cells (GrisP, Porto, Portugal), following the manufactures’ instructions. RNA concentration and sample purity and quality were analyzed using spectrophotometry, as previously described. 1000 ng of RNA was converted to cDNA using Reverse Transcriptase following DNAse I and EDTA treatments. Samples were amplified and converted using the SimplyAmp Thermal Cycler (Applied Biosystems, Life Technologies, Thermo Fisher, Waltham, MA, USA) at 25 °C for 10 min, 42 °C for 60 min, and 70 °C for 10 min.

#### 4.4.4. Real-Time Quantitative Polymerase Chain Reaction

mRNA evaluation of USP42 expression (probe: Hs.PT.58.670189; Integrated DNA Technologies–Integrated DNA Technologies IDT, Leuven, Belgium) was performed using TaqMan^®^ Universal PCR Master Mix-No AmpErase^®^ (ThermoFisher Scientific, Waltham, MA, USA) using TATA-binding protein (TBP) as an endogenous control (probe: Hs.PT.39a.22214825; Integrated DNA Technologies IDT, Leuven, Belgium). The reaction was performed in a MicroAmp™ Optical 96-Well Reaction Plate (ThermoFisher Scientific, Waltham, MA, USA) in a QuantStudio5 machine (Applied Biosystems, Life Technologies, Thermo Fisher, Waltham, MA, USA). Triplicates and non-template controls were performed for all samples. Quantification of mRNA expression was performed using the 2^−ΔΔCt^ method.

#### 4.4.5. Protein Extraction

Cell lysis was performed by incubating the cell pellet for 15 min with RIPA buffer mixed with phosphatase and protease inhibitors. After centrifugation for 10 min at 14,000 rpm at 4 °C, the pellet was discarded. Protein quantification was performed in a 96-well plate using the Modified Protein Lowry assay according to the manufacturers’ instructions. Sample absorbance was read at 650 nm on a SpectraMax^®^ iD3 machine(Molecular Devices, San Jose, CA, USA).

#### 4.4.6. Western Blot Analysis

Whole protein lysates (15 μg) were loaded on 10% polyacrylamide gels. The following antibodies were used: monoclonal USP42 (1:1000, d-4 mouse, sc-390604, Santa Cruz Biotechnology, Dallas, Texas, USA), monoclonal p53 (1:3000, mouse, NCL-L-p53-DO7, Leica Biosystems, Leica, Chicago, IL, USA), monoclonal Cyclin D1 (1:10000, ab134175, Abcam, Cambridge, UK), monoclonal p21 (1:500, rabbit, #2947s, Waf1/Cip1 (12D1), Cell Signaling Technology, Beverly, MA, USA), monoclonal p27 (1:500, rabbit, Kip1–D69C12, #36865, Cell Signaling Technology, Danvers, MA, USA), monoclonal β-actin (1:1000, mouse, Santa Cruz Biotechnology, Dallas, TX, USA), monoclonal α-Tubulin (1:5000, Mouse, T5168, Merck KGaA, Darmstadt, Germany), monoclonal Vinculin (1:3000, Mouse, V9131 Merck KGaA, Darmstadt, Germany), and monoclonal GAPDH (1:1000, Mouse, sc-32233 Santa Cruz Biotechnology, Dallas, TX, USA). Protein expression was analyzed in an Odyssey ^®^ CLx Infrared Imaging System (LI-COR Biosciences, Cambridge, UK) and read at 800 nm absorbance using fluorescent IRDye^®^ 800CW Donkey Anti-Rabbit antibody (LI-COR Biotech, USA) and IRDye^®^ 800CW Goat Anti-Mouse antibody (LI-COR Biotech, Lincoln, NE, USA) as secondary antibodies. Target protein quantification was performed and compared against the total protein of the sample (Ponceau S staining) using Image Studio™ Lite software version 5.2.5 (LI-COR Biosciences, Cambridge, UK).

#### 4.4.7. Phalloidin Staining

Cells were grown in sterilized round slides and transfected with siRNAs, as previously described, before being stained with “Flash Phalloidin™ Green 488” (Biolegend, San Diego, CA, USA) (1:200) and DAPI (1:1000) as previously described [49]. Samples were photographed using the fluorescence microscope Zeiss Axio Imager Z1 (Carl Zeiss, Oberkochen, Germany). Data was analyzed using Image J Version 1.53t software (National Institutes of Health, Bethesda, MD, USA).

#### 4.4.8. Annexin V Assay

siRNA-transfected cells were collected, centrifuged at 300× *g* for 5 min, and resuspended in Binding Buffer 1× (ThermoFisher Scientific, Waltham, MA, USA). Cells were stained with 5 μL Annexin V FITC (ImmunoTools, Friesoythe, Germany) (30 min incubation in the dark) and 50 μg/mL propidium iodate (PI) (5 min incubation in the dark). Unstained cells were used as autofluorescence controls. Cells were filtered (70 μm filter) and read in the BD Accuri C6 cytometer (BD Biosciences, Qume Drive San Jose, CA, USA). At least 20,000 events per sample were evaluated using FlowJo (FlowJo™ Software version 10.9, BD Biosciences, Qume Drive San Jose, CA, USA).

#### 4.4.9. Statistical Analysis

Data analyses were performed using GraphPad Prism (version 8.0.2, La Jolla, CA, USA). Two-way ANOVA or mixed-effect model analysis (non-parametric two-way ANOVA) was performed when applicable. Results are presented as the respective mean and standard error. Samples were considered significantly different when *p* < 0.05 (* *p* ≤ 0.05; ** *p* ≤ 0.01; *** *p* ≤ 0.001; **** *p* ≤ 0.0001) and confidence interval ≠ 1.

## 5. Conclusions

We described a germline *USP42*
*p.(Gly486Arg)* mutation as an underlying cause of FNMTC in a family presenting OSPTC in all the elements under study. Our in silico results support that USPs, other than *USP42*, are altered in sporadic thyroid cancer, and that USPs can be in line with other mutations associated with thyroid cancer development, such as *BRAF p.(Val600Glu)*. Our in vitro results demonstrate that silencing of the *USP42* gene leads to changes in thyroid cell biology. This is shown by differences in cell proliferation, apoptotic rate, and cell cycle-related protein expression, which correlated with the altered protein expression observed in the patients’ tumor samples. We hereby present evidence that *USP42* gene alterations could represent a new susceptibility factor for NSFNMTC.

## Figures and Tables

**Figure 1 ijms-25-01522-f001:**
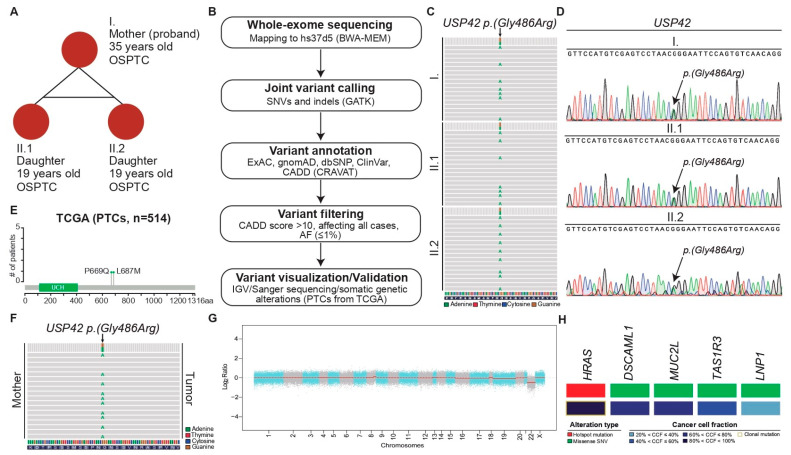
Case description and in silico analysis of non-syndromic FNMTC patients subjected to Whole-Exome Sequencing. (**A**) Family composed of a mother and two monozygotic twin daughters diagnosed with Oncocytic Subtype of Papillary Thyroid Carcinoma (OSPTC). Diagnosis at young age in the three elements of the family and in the absence of syndromic forms of the disease, falling into the category of Non-Syndromic Familial Non-Medullary Thyroid Carcinoma (NSFNMTC). (**B**) Sequencing analysis strategy for the detection of rare, potentially pathogenic Single-Nucleotide Polymorphisms (SNPs) affecting the three individuals included in this study. (**C**) Sequencing reads in Integrative Genomics Viewer showing the *USP42*
*p.(Gly486Arg)* variant detected in the three individuals. (**D**) Representative Sanger electropherograms showing the G > A alteration in the normal DNA taken from the mother (top) and the twin daughters (middle and bottom). (**E**) Lollipop plots depicting the frequencies of the identified somatic mutations affecting *USP42* in 514 Papillary Thyroid Carcinomas subjected to Whole-Exome Sequencing that were selected from the TCGA database. UCH—ubiquitin carboxyl-hydrolase. (**F**) Sequencing reads in Integrative Genomics Viewer showing the *USP42*
*p.(Gly486Arg)* variant detected in the tumor DNA of the mother. (**G**) Copy number plot of the tumor DNA of the mother. The log-ratios are plotted on the y-axis according to genomic positions (x-axis). Chromosomes are depicted by alternating blue and red bands. (**H**) Heatmaps depicting non-synonymous somatic mutations (top) and cancer cell fractions (bottom) in the tumor DNA of the mother. Mutations and cancer cell fractions are color coded according to the legend.

**Figure 2 ijms-25-01522-f002:**
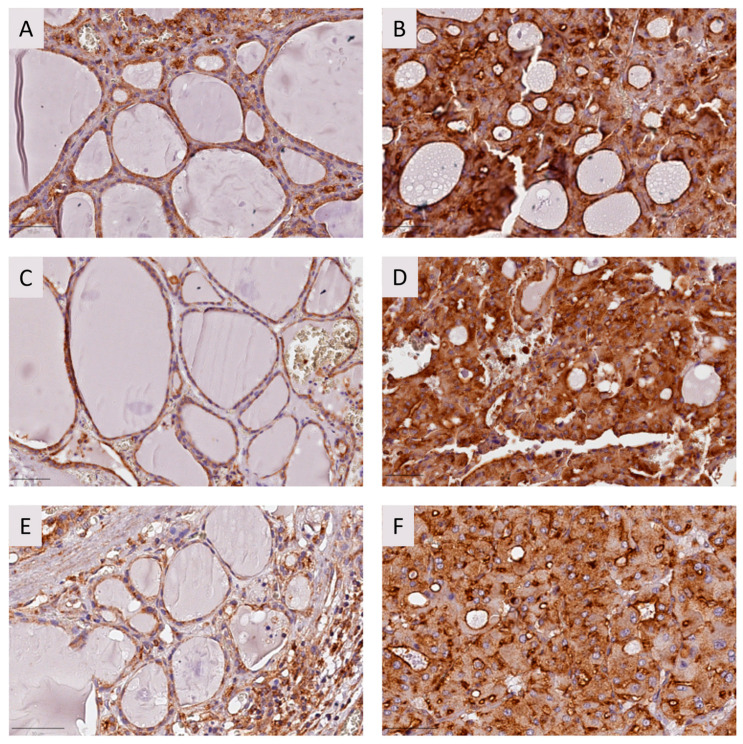
USP42 analysis by immunohistochemistry for FFPE tissues from the affected individuals. Adjacent (**A**) and tumor tissue (**B**) from the proband (I); adjacent (**C**) and tumor tissue (**D**) from twin II.1; and adjacent (**E**) and tumor tissue (**F**) from twin II.2. USP42 is overexpressed in the analyzed tumor tissues compared to adjacent tissue in the three individuals. Staining intensity was evaluated by semi-quantitative analysis. Bar: 50 μm.

**Figure 3 ijms-25-01522-f003:**
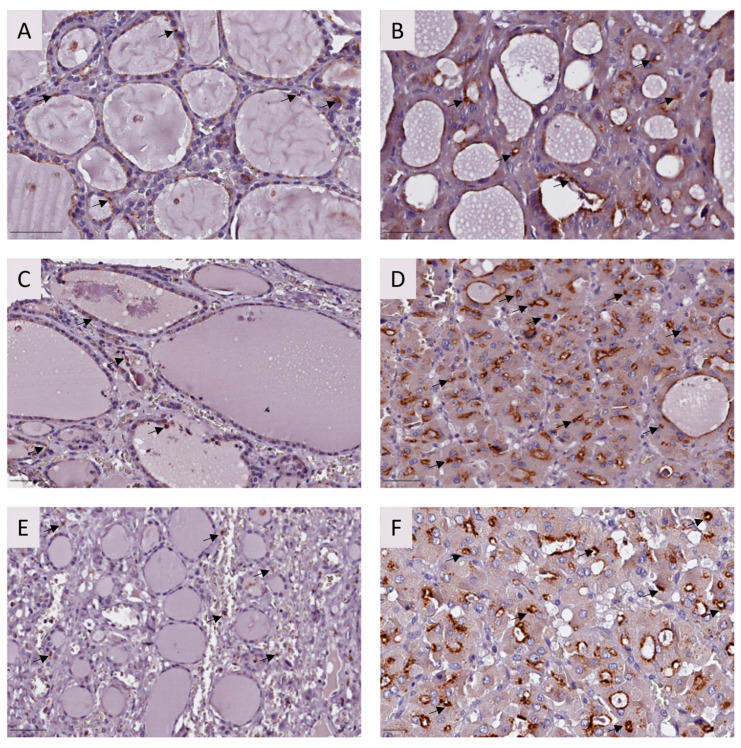
Caspase-3 analysis by immunohistochemistry for FFPE tissues from the affected individuals. Adjacent (**A**) and tumor tissue (**B**) from the proband (I); adjacent (**C**) and tumor tissue (**D**) from twin II.1; and adjacent (**E**) and tumor tissue (**F**) from twin II.2. Caspase-3 is overexpressed in the analyzed tumor tissues compared to adjacent tissue in the three individuals. Black arrows indicate positive cells. Staining intensity was evaluated by semi-quantitative analysis. Bar: 50 μm.

**Figure 4 ijms-25-01522-f004:**
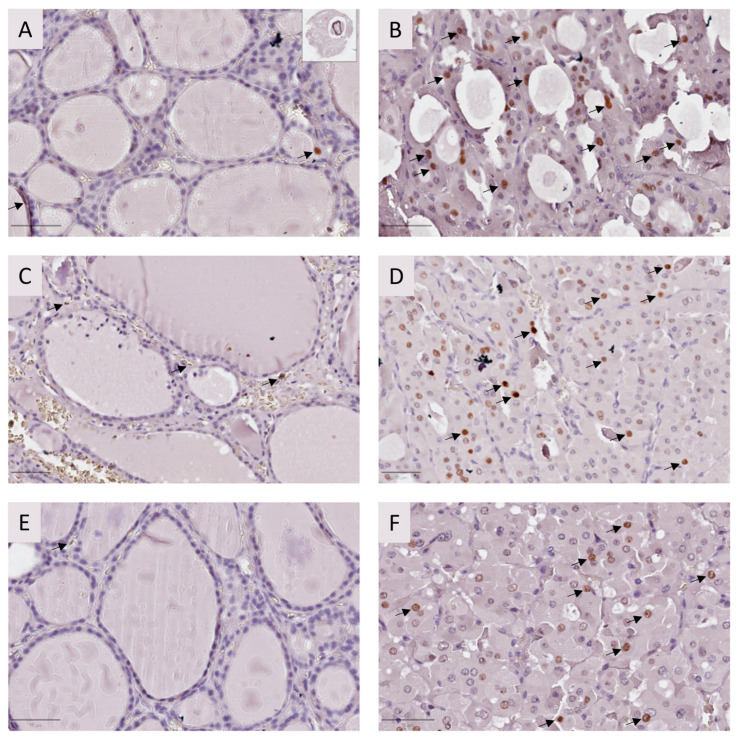
p53 analysis by immunohistochemistry for FFPE tissues from the affected individuals. Adjacent (**A**) and tumor tissue (**B**) from the proband (I); adjacent (**C**) and tumor tissue (**D**) from twin II.1; and adjacent (**E**) and tumor tissue (**F**) from twin II.2. p53 is overexpressed in the analyzed tumor tissues compared to adjacent tissue in the three individuals. Black arrows indicate positive cells. Bar: 50 μm.

**Figure 5 ijms-25-01522-f005:**
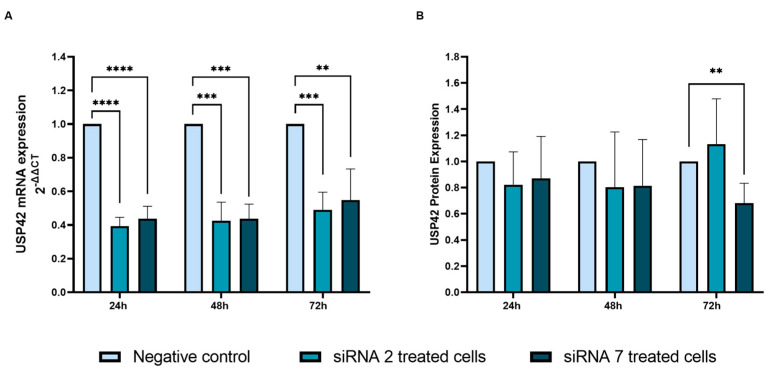
(**A**) USP42 mRNA expression in Nthy-Ori 3-1 cells after siRNA treatment. USP42 mRNA expression was decreased with a maximal 60% silencing effect in the Nthy-Ori 3-1 cell line using siRNA treatments. (**B**) USP42 protein expression following siRNA treatment. Protein silencing was significant at the 72 h timepoint in the Nthy-Ori 3-1 cell line following siRNA 7 treatment (*p* = 0.008). Each experiment was performed with two replicates and three independent experiments were conducted (** *p* ≤ 0.01; *** *p* ≤ 0.001; **** *p* ≤ 0.0001).

**Figure 6 ijms-25-01522-f006:**
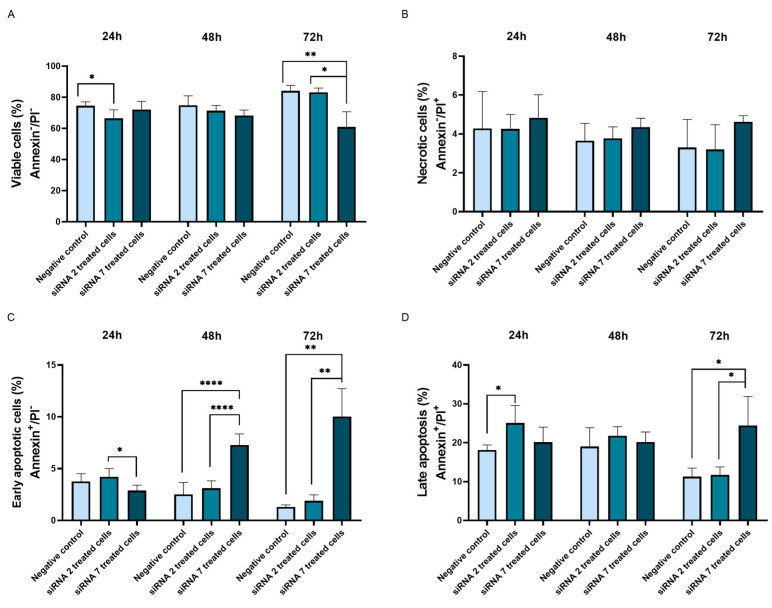
Cell viability and cell mortality analysis at the studied timepoints. (**A**) Cell viability percentage of Annexin V^−^/PI^−^ stained cells. (**B**) Percentage of necrotic cells (Annexin V^−^/PI^+^); (**C**) Percentage of early apoptotic cells (AnnexinV^+^/PI^−^); (**D**) Percentage of late apoptotic/post-apoptotic necrotic cells (Annexin V^+^/PI^+^). Each experiment was performed in two replicates, and three independent experiments were conducted (* *p* ≤ 0.05; ** *p* ≤ 0.01; **** *p* ≤ 0.0001). PI-Propidium iodate.

**Figure 7 ijms-25-01522-f007:**
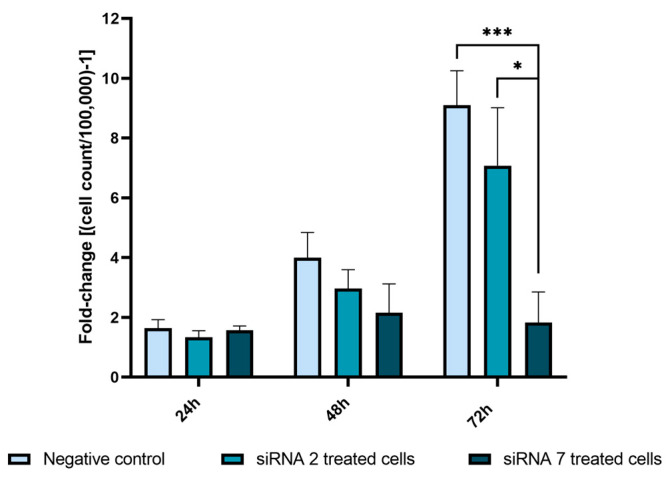
Nthy-Ori 3-1 cell proliferation analysis following *USP42* silencing treatment. Two to three cell counts were performed per well to establish cell proliferation rates. Fold-change was calculated from the initial number of plated cells [cell count/100,000)-1]. This evaluation was conducted using two replicates and two independent experiments were conducted (* *p* ≤ 0.05; *** *p* ≤ 0.001).

**Figure 8 ijms-25-01522-f008:**
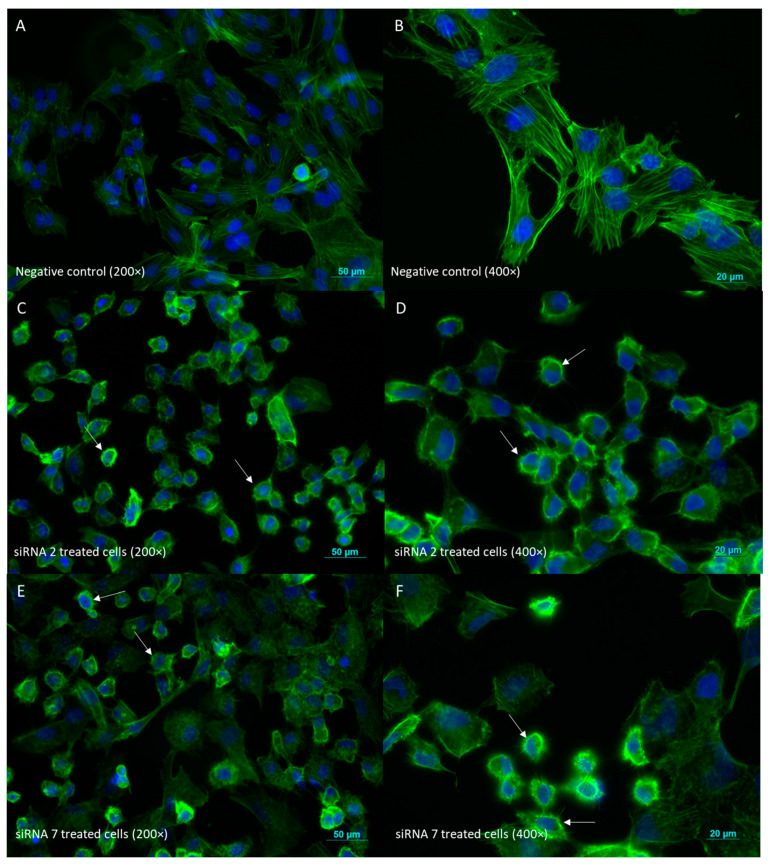
Immunofluorescence analysis of cell morphology by phalloidin staining of actin filaments (green) contrasted with DAPI staining of cell nuclei (blue). (**A**,**B**) Negative control cells under 200× and 400× magnification, respectively. (**C**,**D**) siRNA 2-treated cells under 200× and 400× magnification, respectively. (**E**,**F**) siRNA 7-treated cells under 200× and 400× magnification, respectively. White arrows indicate cells with altered morphology, presenting loss of shape and actin concentrated in cell periphery. Images are representative of three experiments conducted independently (each with two replicates). Bar: 50 μm at 200× magnification and 20 μm at 400× magnification.

**Figure 9 ijms-25-01522-f009:**
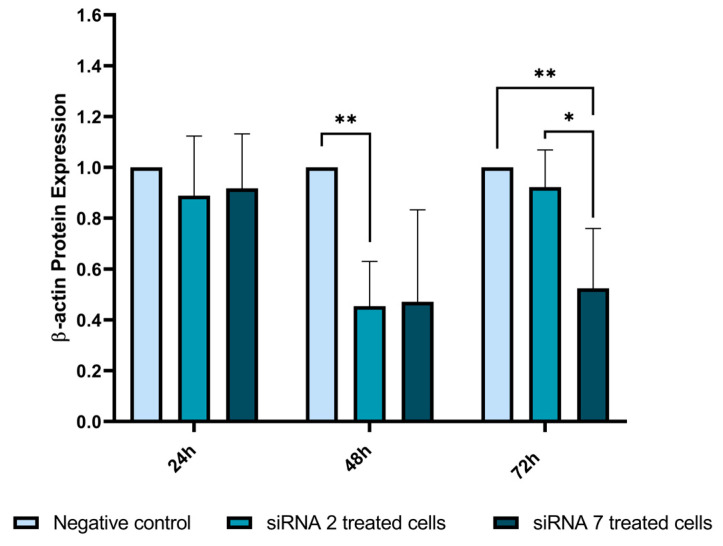
β-actin protein expression following siRNA treatment. siRNA 2-treated cells showed a 55% reduction in β-actin expression at 48 h (*p* = 0.002) and siRNA 7-treated cells showed a 40% reduction 72 h post-treatment (*p* = 0.010). Each experiment was performed with two replicates and three independent experiments were conducted. (* *p* ≤ 0.05; ** *p* ≤ 0.01).

**Figure 10 ijms-25-01522-f010:**
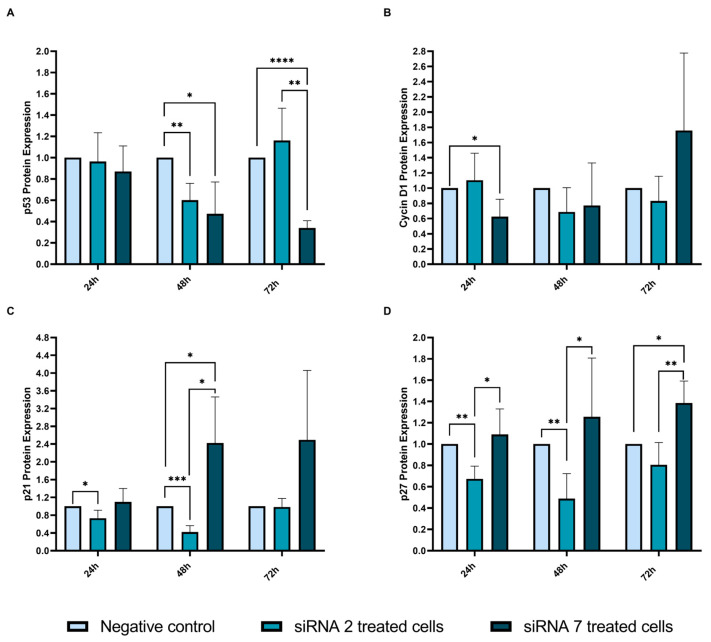
Cell cycle-related protein expression following siRNA treatment of Nthy-Ori 3-1 cells. (**A**) p53 protein expression; (**B**) Cyclin D1 protein expression; (**C**) p21 protein expression; and (**D**) p27 protein expression. Each experiment was performed with two replicates and three independent experiments were conducted (* *p* ≤ 0.05; ** *p* ≤ 0.01, *** *p* ≤ 0.001, **** *p* ≤ 0.0001).

**Table 1 ijms-25-01522-t001:** Candidate SNP identified in the three affected individuals included in this study.

^1^ ID	Position (^2^ Chr. 7)	Alternate Allele	Gene	Mutation Type	Amino Acid Change	gnomAD AF Total	dbSNP	CADD	ClinVar
I	6189283	G > A	*USP42*	Missense	*p.(Gly486Arg)*	0.0004582	rs200908439	23.6	^3^ US
II.1	6189283	G > A	*USP42*	Missense	*p.(Gly486Arg)*	0.0004582	rs200908439	23.6	US
II.2	6189283	G > A	*USP42*	Missense	*p.(Gly486Arg)*	0.0004582	rs200908439	23.6	US

^1^ ID—Identification; ^2^ Chr.—Chromosome; ^3^ US—Uncertain Significance.

**Table 2 ijms-25-01522-t002:** Somatic mutations detected in the tumor DNA subjected to Whole-Exome Sequencing.

^1^ ID	^2^ Chr.	Position	Alternate Allele	Gene	Mutation Type	Amino Acid Change	Alt. Reads	Ref. Reads	Tumor ^3^ MAF	Tumor Depth	Cancer Cell Fraction	Clonal Status
I	11	533874	T > C	*HRAS*	Missense	*p.(Gln61Arg)*	171	68	0.2845	239	0.93	Clonal
I	3	100170600	A > T	*LNP1*	Missense	*p.(His78Leu)*	37	5	0.119	42	0.39	Subclonal
I	1	1269225	G > A	*TAS1R3*	Missense	*p.(Cys647Tyr)*	165	25	0.1316	190	0.43	Subclonal
I	6	30954414	C > G	*MUC21*	Missense	*p.(Asp154Glu)*	39	9	0.1875	48	0.61	Subclonal
I	11	117299237	G > A	*DSCAML1*	Missense	*p.(Pro2050Leu)*	24	7	0.2258	31	0.74	Subclonal

^1^ ID—Identification; ^2^ Chr.—Chromosome; ^3^ MAF— Minor Allele Frequency.

**Table 3 ijms-25-01522-t003:** p53 quantification in different lesions from the three family elements.

^1^ ID	Sample	Lesion	Total (N)	p53 (+)	p53 (−)	Ratio (%)
I	Sample 1	Tumor	1084	624	460	57.56
		Adjacent tissue	1026	67	959	6.53
	Sample 2	^2^ FND	1069	29	1040	2.71
II.1	Sample 3	Tumor	1092	453	639	41.48
		Adjacent tissue	1005	107	898	10.65
	Sample 4	Tumor	1094	288	806	26.33
		Adjacent tissue	1071	8	1063	0.75
II.2	Sample 5	Tumor	1006	536	470	53.28
		Adjacent tissue	1017	14	1003	1.38
	Sample 6	Tumor	902	431	471	47.78
		Adjacent tissue	1027	48	979	4.67

At least 1000 cells were counted in each field. Quantification was performed in the areas that presented higher expression of p53. ^1^ ID—Identification; ^2^ FND—Follicular Nodular Disease.

## Data Availability

Dataset from The Cancer Genome Atlas (TCGA) analyses can be accessed/download in cBioPortal for Cancer Genomics at https://www.cbioportal.org/ (accessed on 15 November 2023).

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
