# Peer review of "Investigating USP42 Mutation as Underlying Cause of Familial Non-Medullary Thyroid Carcinoma"

_ijms, 2024, doi:10.3390/ijms25031522_

Round 1

Reviewer 1 Report

Comments and Suggestions for Authors

The authors describe a pathogenic mutation in the USP42 (p.Gly486Arg) gene in three members of a family with non-syndromic FNMTC. The authors then perform bioinformatic and in vivo testing of the causative pathogenic role of this mutation. The manuscript is well-written, and the experiments and their results are clearly explained.

The authors mention that the histopathology revealed an oncocytic variant of PTC. But as the tumors also had a somatic HRAS pathogenic variant, it may to an extent, explain the possible subtype of this PTC. It is likely that this oncocytic variant of PTC, may have had a follicular architecture suggestive of a RAS pathogenic variant (as opposed to the classic papillary architecture, which is mainly BRAF-driven, and BRAF and RAS are mutually exclusive molecular alterations in differentiated thyroid cancers). So, it would be interesting for the authors to check and clarify whether the histopathology revealed a papillary architecture or follicular architecture of cells (the latter would further support the involvement of RAS pathway).

In section 4.3.4, what is the expansion of CCF?

Comments on the Quality of English Language

Minor improvements are necessary.

Reviewer 2 Report

Comments and Suggestions for Authors

The authors investigate the role of USP42 in familial non-medullary thyroid carcinoma. The manuscript is very well-written. Some concerns include:

1. Incorporating representative histopathology images of the cases will ensure a better understanding of the work

2. Interpretation, evaluation and scoring used for caspase 3 and USP42 immunohistochemistry results have not been detailed. As part of legend to figure 2, the authors have added the following statement: "USP42 is overexpressed in the analyzed tumor tissues when comparing to adjacent tissue in the three individuals." This is not very evident from this image. If done subjectively, it may be a false interpretation just because of the crowded nature of cells in the tumor tissues as compared to adjacent normal. Hence, method of interpretation and scoring need to be detailed, and if possible statistical analysis performed.

Moreover, it will be better if IHC results in thyroid tissues from unrelated subjects can also be documented. These samples, considering the hypothesis, are expected to show a lower level of protein expression.

3. GAPDH, Vinculin and tubulin were also processed for western blot analysis but the details of their antibodies have not been provided in methods. Also, the authors state in line 275 that "We also detected a reduction of tubulin and 274 vinculin expression by Western blot (Figure S3)." this needs further detailing in the form of concentrations obtained and statistical analysis. 

4. Lines 121-123: "When looking at mutations affecting the USP42 gene in 514 PTCs subjected to WES taken from TCGA, two missense mutations were identified, with one of them being pathogenic (Figure 1E; Figure S1; Table S1)." Although Figure 1E does depict two variants of USP42, Figure S1 and Table S1 do not depict the same. It will also be preferred as to mention in the text which one of the two was pathogenic.

5. Some of the text incorporated in Figure 1 is too small to be read.
